# Development of Attenuated Viruses for Effective Protection against Pepper Veinal Mottle Virus in Tomato Crops

**DOI:** 10.3390/v16050687

**Published:** 2024-04-26

**Authors:** Guan-Da Wang, Chian-Chi Lin, Tsung-Chi Chen

**Affiliations:** Department of Medical Laboratory Science and Biotechnology, Asia University, Wufeng, Taichung 41354, Taiwan; denny94207z@gmail.com (G.-D.W.); m65391721@gmail.com (C.-C.L.)

**Keywords:** cross-protection, mild virus strain, Solanaceae crops, *Potyvirus*

## Abstract

Tomato (*Solanum lycopersicum*) is the most important vegetable and fruit crop in the family Solanaceae worldwide. Numerous pests and pathogens, especially viruses, severely affect tomato production, causing immeasurable market losses. In Taiwan, the cultivation of tomato crops is mainly threatened by insect-borne viruses, among which pepper veinal mottle virus (PVMV) is one of the most prevalent. PVMV is a member of the genus *Potyvirus* of the family *Potyviridae* and is non-persistently transmitted by aphids. Its infection significantly reduces tomato fruit yield and quality. So far, no PVMV-resistant tomato lines are available. In this study, we performed nitrite-induced mutagenesis of the PVMV tomato isolate Tn to generate attenuated PVMV mutants. PVMV Tn causes necrotic lesions in *Chenopodium quinoa* leaves and severe mosaic and wilting in *Nicotiana benthamiana* plants. After nitrite treatment, three attenuated PVMV mutants, m4-8, m10-1, and m10-11, were selected while inducing milder responses to *C. quinoa* and *N. benthamiana* with lower accumulation in tomato plants. In greenhouse tests, the three mutants showed different degrees of cross-protection against wild-type PVMV Tn. m4-8 showed the highest protective efficacy against PVMV Tn in *N. benthamiana* and tomato plants, 100% and 97.9%, respectively. A whole-genome sequence comparison of PVMV Tn and m4-8 revealed that 20 nucleotide substitutions occurred in the m4-8 genome, resulting in 18 amino acid changes. Our results suggest that m4-8 has excellent potential to protect tomato crops from PVMV. The application of m4-8 in protecting other Solanaceae crops, such as peppers, will be studied in the future.

## 1. Introduction

Tomato (*Solanum lycopersicum*) is one of the most economically important species in the family Solanaceae and the ninth most cultivated agricultural crop globally [1,2]. Based on the Food and Agriculture Organization (FAO) report of the United Nations http://www.fao.org (accessed on 21 December 2021), global tomato crops had an economic output value reaching USD 19.5 billion in 2021. Tomato is also an important crop in Taiwan, with a total planting area of about 4100 ha and a total yield of 98,340 tons, which contributed to USD 141 million output value in 2021 (Agriculture and Food Agency, Council of Agriculture, Executive Yuan, https://www.afa.gov.tw/ (accessed on 29 July 2022)).

Tomato cultivation in Taiwan suffers from numerous insect-borne viruses, including potato virus Y (PVY) [3], chilli veinal mottle virus (ChiVMV) [4], pepper veinal mottle virus (PVMV) [5], pepper mottle virus (PepMoV) [6], and tobacco vein banding mosaic virus (TVBMV) [7] of the aphid-transmitted genus *Potyvirus* in the family *Potyviridae*. Viruses belonging to the genus *Potyvirus* have a monopartite, single-stranded positive-sense RNA genome of about 10 kb, encapsidated by multiple units of a single type of coat protein (CP) in flexuous and filamentous particles of 720–850 nm long and 12–15 nm in diameter. The 5′ end of potyviral genomic RNA is covalently linked with a viral protein genome-linked (VPg) and its 3′ terminus with a polyadenylation [8]. Potyviral genomic RNA contains a large open reading frame (ORF) encoding a polyprotein. The polyprotein is proteolytically processed by viral-encoded proteinases, P1, helper component proteinase (HC-Pro), and nuclear inclusion a-protease (NIa-Pro), to produce 10 mature products, P1, HC-Pro, P3, 6K1, cylindrical inclusion protein (CI), 6K2, VPg, NIa-Pro, nuclear inclusion b (NIb), and CP [8]. P1 and HC-Pro are essential for potyvirus infection by facilitating virus movement and acting as viral suppressors of RNA silencing (VSRs) to counteract the host plant RNA silencing defensive response [9,10,11]. In addition to the large polyprotein, a *trans*-frame product, P3N-PIPO, sharing the N-terminal region of P3, is generated via transcriptional slippage during viral replication and is responsible for cell-to-cell movement [12,13,14]. Potyviruses can be efficiently transmitted by aphids in a nonpersistent manner. CP plays a role in the attachment of virions to aphid stylets. HC-Pro facilitates the binding between CP and aphid receptors in aphid stylets [15].

Although pesticides and shelter have been used to protect crops from insect-borne virus invasion, triggering gene silencing in plants is the most effective measure for controlling viral diseases. Genetic modification via the insertion of viral genomic fragments into the nuclear chromosomes of plants for inducing post-transcriptional gene silencing (PTGS) in plants has been exploited successfully for generating resistance to various viruses, including potyviruses. For example, transgenic papaya lines expressing the CP gene of papaya ringspot virus (PRSV) were generated successfully against PRSV [16]. The transgenic squash line ZW-20H, carrying integrated CP genes of zucchini yellow mosaic virus (ZYMV) and watermelon mosaic virus (WMV), is highly resistant to mixed infections with ZYMV and WMV [17]. Virus-induced gene silencing (VIGS) using viral vector technology is an alternative approach to inducing RNA silencing-mediated defense in infected plants. Double-stranded RNA (dsRNA) replicative intermediates of viral vectors are conducive to triggering RNA silencing against their inserted counterparts [18,19]. The exogenous application of dsRNA molecules has also been shown to induce virus resistance [20]. Topical spraying of dsRNA molecules homologous to PVY CP genes induced significant and specific resistance against PVY in tomato plants [21].

PTGS-induced genetically modified (GM) crops effectively resist virus invasion; however, their application is prohibited in many areas due to public food safety and ecological concerns. Mild virus strain-induced cross-protection provides an alternative approach to virus resistance. Cross-protection is based on a natural phenomenon in which the tolerance or resistance of a plant to an aggressive virus strain is induced by systemic infection with a relatively mild virus [22,23]. Protective mild virus strains have been successfully used in the large-scale control of viral diseases in Taiwan, such as the natural mild strain of ZYMV, WK, applied to cucurbit crops [24] and the nitrous acid-induced strain PRSV HA5-1 applied to papaya [25,26]. Viruses may spontaneously mutate during replication, reducing their virulence. However, the natural mutation rate is low, and attenuated viruses are overshadowed by aggressive viruses in nature. Under artificial conditions, treatments with physical agents, such as ultraviolet light (UV) and temperature changes, and chemical mutagens, such as nitrite, significantly increase the probability of attenuating viruses [27]. Cross-protection is a cross-talk of PTGS, protein-mediated resistance, and plant innate immunity, such as salicylic acid signaling a defense reaction [28,29,30]. A genome sequence analysis of protective mild virus strains showed that attenuation was associated with point mutations in their VSRs [27]. Therefore, protective mild viruses can be generated by constructing dysfunctional VSRs [28,31].

PVMV was first found in Taiwan in 2006 [5] and has become the most prevalent potyvirus infecting tomato and pepper crops in Taiwan [32]. Previous reports have shown that nitrite treatment is suitable for the attenuation of potyviruses [25,33]. Furthermore, protective mild potyviruses can be selected based on the criteria that the virus can infect *Chenopodium quinoa* but fail to cause local lesions in inoculated leaves [28,31,33]. In this study, we attempted to attenuate PVMV using nitrous acid treatment and select its mild strains in *C. quinoa*. The protective effect of mild strains of PVMV on tomato plants was also demonstrated.

## 2. Materials and Methods

### 2.1. Virus Source

The PVMV tomato isolate Tn collected in Tainan City of southern Taiwan, was used for mutagenesis. Another PVMV isolate, Xs-1, collected from sweet pepper in Xinyi Township, Nantou County of central Taiwan, was used as an alternative challenger. Both were isolated from single lesions in inoculated leaves of *C. quinoa* and maintained in the indicator plants *C. quinoa* and *Nicotiana benthamiana* and the natural host tomato (*Solanum lycopersicum*) using mechanical inoculation, as previously described [34]. Crude sap from virus-infected leaves homogenized in inoculation buffer (10 mM K_2_HPO_4_, 10 mM KH_2_PO_4_, and 10 mM Na_2_SO_3_, pH 7.0) was used as the inoculum. Virus-infected plants were grown in a temperate-controlled (27 °C) greenhouse and a temperature-controlled (27 °C) plant growth chamber with a light:dark photoperiod of 16:8 h.

### 2.2. Nitrite Treatment

Nitrite-induced mutagenesis was conducted as previously described [25], with modifications. Briefly, PVMV Tn-infected *N. benthamiana* leaves were ground in distilled water (3 g tissue in 10 mL water), and the lysate was filtrated using gauze. Crude sap was centrifuged at 7155× *g* for 10 min using a GRF-L-50-6 rotor (LaboGene, Lillerød, Denmark). The supernatant was collected and treated with a 5 mL solution consisting of 400 mM sodium nitrite (NaNO_2_) and 100 mM sodium acetate (NaOAc) at a pH of 6.0, incubating at 37 °C for 30–45 min. The reaction was stopped by adding an equal volume of 10 mM potassium phosphate buffer (10 mM K_2_HPO_4_, 10 mM KH_2_PO_4_, and 10 mM Na_2_SO_3_, pH 8.0). The supernatant treated with sterile deionized water was used as a control. The mixture was mechanically inoculated, with a cotton pad, to *C. quinoa* leaves dusted with 600 mesh carborundum. The inoculated plants were kept in a temperature-controlled (27 °C) plant growth chamber with a 16:8 h light:dark photoperiod.

### 2.3. Screening of Attenuated Mutants

Leaves of *C. quinoa* inoculated with nitrite-treated virus solution were examined using a Multigel-21 imaging system (TOPBIO, New Taipei City, Taiwan) at day 7 post-inoculation under UV light with a wavelength of 365 nm. Fluorescence was visualized using a red filter (600 nm wavelength) to monitor local lesion formation. Fluorescent spots that were observed under UV light but did not cause necrotic lesions in *C. quinoa* leaves were picked and used as inocula and transferred to another *C. quinoa* leaf for isolation. Putative attenuated mutants were transferred to the systemic hosts *N. benthamiana* and tomato line SV-055 (Known-You Seed Co., Ltd., Kaohsiung, Taiwan) plants to verify their infectivity and pathogenicity. The test plants were kept in a plant growth chamber and a greenhouse, as mentioned above, for symptom observation for at least 3 weeks. Virus infection and titers were monitored with an indirect enzyme-linked immunosorbent assay (ELISA) using the antiserum against the PVMV CP (RAs PVMV-CP) [34] previously produced in our laboratory.

### 2.4. Cross-Protection Evaluation

*N. benthamiana* plants at the three- to four-true leaf stage and tomato seedlings at the two-true leaf stage were mechanically inoculated with attenuated PVMV mutants. After 4 days of inoculation with mild viruses, test *N. benthamiana* plants were challenged with severe viruses, the wild-type (WT) PVMV Tn or the sweet pepper isolate Xs-1, using mechanical inoculation on uninoculated upper leaves. Tomato plants were challenged with PVMV Tn 3 to 14 days after inoculation with mild viruses. The protection effects were evaluated based on symptom severity. The disease severity index (DSI), with the formula DSI (%) = $\frac{\Sigma \;(\mathrm{Class}\;\mathrm{frequency}\;\times \;\mathrm{score}\;\mathrm{of}\;\mathrm{rating}\;\mathrm{class})}{\left(\mathrm{Total}\;\mathrm{number}\;\mathrm{of}\;\mathrm{observations})\;\times \;(\mathrm{maximal}\;\mathrm{disease}\;\mathrm{index}\right)}$ × 100 [35], was calculated to express virus impact. Symptom development was ranked in arbitrary scores for calculation. In addition, bioassays that inoculated the crude sap of test plants to *C. quinoa* leaves were also performed to differentiate infections with mild or aggressive viruses. No local lesions produced were judged to be protective. The protective efficacy (PE) of mild virus strains was calculated as PE = [Infection rate (IR) in the untreated population (IRU) − IR in the protected population (IRP)/IRU] × 100%, where IR = (number of PVMV Tn infection/total number of test plants) × 100% [36]. Significant differences in treatments were analyzed using the Mann–Whitney *U* test [37].

### 2.5. Indirect ELISA

Three leaf disks (0.5 cm in diameter) from each plant were collected by punching for assay. ELISA plates were coated with crude leaf sap at a 1/100 dilution in coating buffer (15 mM Na_2_CO_3_, 34 mM NaHCO_3_, and 3 mM NaN_3_, pH 9.6). Aliquots of 200 μL were loaded to each well. The plates were incubated at 37 °C for 30 min, then washed with PBST buffer (136 mM NaCl, 1 mM KH_2_PO_4_, 8 mM Na_2_HPO_4_·12H_2_O, 2 mM KCl, 3 mM NaN_3_ and 0.05% Tween 20) three times, each for 3 min. RAs PVMV-CP [34] was diluted with conjugate buffer (PBST buffer containing 2% PVP-40 and 0.2% ovalbumin) at a 1/2000 dilution and loaded into plate wells, with 200 μL in each well. The plates were incubated at 37 °C for 30 min and then washed three times with PBST, each for 3 min. The alkaline phosphatase (AP)-conjugated goat anti-rabbit IgG (Jackson ImmunoResearch, West Grove, PA, USA) was diluted at a 1/5000 dilution in conjugate buffer, and aliquots of 200 μL were loaded to each well. The plates were incubated at 37 °C for 30 min, and then the wash step was repeated as previously described. The color-developing solution was prepared by dissolving ρ-nitrophenyl phosphate disodium hexahydrate (ρ-NPP) in substrate buffer (9.7% diethanolamine and 3 mM NaN_3_, pH 9.8) to a final concentration of 1 mg/mL, and 180 μL of the solution was loaded into each well. The absorbance at 405 nm was recorded using a Model 680 microplate reader (Bio-Rad, Hercules, CA, USA) for 1 h after adding the enzyme substrate.

### 2.6. Viral Whole-Genome Sequencing

Total RNA was extracted from virus-infected *N. benthamiana* leaf tissue using a Plant Total RNA Miniprep Purification Kit (GMbiolab, Taichung, Taiwan) following the manufacturer’s instructions and used as the template. PVMV-specific primer pairs were used to amplify overlapping fragments of the viral genome (Appendix A). The SuperScrip IV reverse transcriptase system (ThermoFisher, Waltham, MA, USA) was used for the synthesis of cDNA using 1 μg of total RNA as the template. All PCR reactions were performed in a Veriti thermocycler (ThermoFisher). Each PCR reaction contained 100 ng of cDNA, 0.5 U of KOD-Plus DNA polymerase (Toyobo, Osaka, Japan), 2.5 μL of 10× buffer, 25 mM MgSO_4_, 2.5 μL of 2 mM dNTP mix, 50 pM of each primer, and nuclease-free water to a final volume of 25 µL. The PCR conditions were set as initial denaturation at 94 °C for 2 min, followed by 35 cycles of 94 °C for 1 min, 58 °C for 2 min, and 72 °C for 4.5 min, and then a final extension step for 7 min at 72 °C. Amplicons were visualized using electrophoresis on a 1% agarose gel stained with SafeView DNA stain (Applied Biological Materials, Richmond, BC, Canada). PCR products were purified from the gel using a Plus DNA Clean/Extraction Kit (GMbiolab), cloned into the pTOPO-XL2 vector (ThermoFisher) following the manufacturer’s instructions, and sequenced by Mission Biotech Inc, Taipei, Taiwan using ABI 3730xl DNA analyzer (ThermoFisher). Three clones were sequenced for each amplicon. The complete viral genome sequence was assembled from contigs.

### 2.7. Sequence Analysis

The genome sequences of different PVMV isolates were downloaded from the National Center for Biotechnology Information (NCBI) GenBank database. Their accession numbers are shown in Appendix A. Pairwise comparisons of genomic sequences of PVMV isolates were performed using NCBI’s basic local alignment search tool (BLAST) (https://blast.ncbi.nlm.nih.gov/Blast.cgi (accessed on 28 November 2023)). Multiple alignments of nucleotide (nt) and amino acid (aa) sequences of the full-length genomes were conducted using the ClustalW program of the MEGA X software vision 10.0.5 (Molecular Evolutionary Genetics Analysis, Philadelphia, PA, USA). Phylogenetic analyses were conducted using the Tree Explorer program of MEGA X, with 1000 bootstrap replicates. Phylogenetic branches were set as the neighbor-joining method.

## 3. Results

### 3.1. Screening of Non-HR PVMV Mutants after Nitrite Treatment

A series of 10 nitrite mutagenesis experiments were conducted to induce mutations in PVMV Tn. The virus solutions, treated with nitrite, were then applied to *C. quinoa* leaves through mechanical rubbing to observe the formation of virus-induced local lesions. At 10 days post-inoculation (dpi), three distinct types of local lesions emerged and were classified as Types I–III. Type I lesions manifested as necrotic spots resembling those caused by the WT PVMV Tn, while Type II lesions exhibited chlorotic spots without necrosis. Type III lesions, displaying a blurred appearance, were easily detectable under UV illumination at a 365 nm wavelength, appearing as bright as Type I and Type II spots (Figure 1A). The total count of local lesions reached 1769, with 1342 categorized as Type I, 415 as Type II, and 12 as Type III (Table 1). Subsequently, the 12 Type III lesions were isolated using standard single-lesion isolation procedures to obtain distinct virus isolates.

The virulence of the 12 virus isolates was evaluated on *N. benthamiana* plants. Two isolates caused severe terminal bud atrophy and plant wilting similar to that caused by WT PVMV Tn. Seven caused leaf curling, mosaic, and stunting, but not wilting. In contrast, three isolates, denoted m4-8, m10-1, and m10-11, exhibited weakly virulent symptoms of leaf curling and blistering (Figure 1B) and were chosen for subsequent experiments.

### 3.2. Assessment of the Protective Effect of the Attenuated Mutants on N. benthamiana Plants

*N. benthamiana* plants were used for assays at the three- to four-true leaf stage. The accumulations of m4-8, m10-1, and m10-11 in both inoculated and systemic (non-inoculated) leaves of *N. benthamiana* plants were monitored using indirect ELISA with RAs PVMV-CP. Crude sap from three leaf disks (0.5 cm in diameter) punched from the leaves of each plant part was used for detection. The assay spanned 14 days from the initiation of inoculation, with detection concluding at 14 dpi due to severe wilting in PVMV Tn-infected plants. A comparative analysis utilized *N. benthamiana* plants inoculated with buffer or PVMV Tn, with four plants tested for each treatment. The results indicated that PVMV Tn was detectable in inoculated leaves at 2 dpi and systemic leaves at 3 dpi. In contrast, m4-8, m10-1, and m10-11 were detected in inoculated and systemic leaves at 8 dpi and 4 dpi, respectively. Notably, no significant differences in accumulation levels were observed in the systemic leaves of *N. benthamiana* plants among PVMV Tn, m4-8, m10-1, and m10-11 (Appendix A).

The protective effects of m4-8, m10-1, and m10-11 on *N. benthamiana* plants were evaluated within a controlled plant growth chamber. Fifteen plants pre-inoculated with each mutant were included in three independent experiments. Following the detection of the three mutants in the systemic leaves of *N. benthamiana*, plants were challenged with WT PVMV Tn on day four of pre-inoculation. As a control, 15 mock-inoculated *N. benthamiana* plants were also subjected to the PVMV Tn challenge. Virus infection was confirmed in all test plants through indirect ELISA. The evaluation of the protective effect was based on disease development over a 14-day period. Additionally, bioassays were conducted, utilizing the formation of local lesion types on *C. quinoa* leaves to distinguish mutants from WT PVMV Tn (Figure 2). The disease severity index (DSI) was determined on an ordinal scale ranging from 0 to 4 (0 = symptomless, 1 = leaf curling and blistering as caused by m4-8, 2 = mosaic and slight stunting, 3 = severe stunting but not wilting, and 4 = wilting or dead as caused by WT PVMV Tn). Throughout the testing period, all m4-8-treated plants showed mild symptoms (scale 1) in *N. benthamiana* plants and did not develop WT PVMV Tn-induced local lesions on *C. quinoa* leaves. In contrast, eight out of fifteen tested plants pre-inoculated with m10-1 or m10-11 displayed severe symptoms (scale 2 or 3) 10 days after PVMV Tn challenge, inducing typical necrotic spots on *C. quinoa* leaves in bioassays. Results showed that m4-8 provided complete protection (100%) against PVMV Tn, while m10-1 and m10-11 exhibited relatively low protective efficacy (46.7%) (Table 2 and Appendix A).

To assess the protective efficacy of m4-8 against a heterologous PVMV source, Xs-1 obtained from sweet pepper in central Taiwan was used as the challenge virus. In two separate experiments, 10 *N. benthamiana* plants were challenged with Xs-1 on the fourth day after pre-inoculation with m4-8. Concurrently, an equal number of mock-inoculated *N. benthamiana* plants were subjected to Xs-1 as the control group. The progression of symptoms on the test plants was recorded for 14 days. Following the symptom observation period, all tested plants underwent bioassays to differentiate between m4-8 and Xs-1 infections. Our results showed that m4-8 also provided complete protection (100%) against Xs-1 (Table 2 and Appendix A).

### 3.3. Mild Strains of PVMV also Protect Tomato Plants

Prior to evaluating the protective efficacy of the PVMV mutants on tomato plants, five tomato varieties purchased from Known-You seed company (Kaohsiung, Taiwan) were examined for their susceptibility to the WT PVMV Tn. Tomato seedlings at the two-true leaf stage were used for the assay. Despite the absence of apparent symptoms in the inoculated plants, virus infection and accumulation were validated through indirect ELISA. Crude saps obtained from three leaf disks (0.5 cm in diameter) punched from uninoculated upper leaves of each plant (*n* = 4) were employed for the assay. The assay was performed weekly for 3 months, using mock-inoculated tomato plants as a comparative reference. Results showed that all tested tomato varieties exhibited susceptibility to PVMV Tn. SV-055, a popular commercial variety in Taiwan, was selected for further investigation. The accumulation of PVMV Tn in tomato SV-055 plants could be detected in the first week after inoculation, increased significantly in the second week, and then decreased until 7 weeks after inoculation (Figure 3).

The protective effectiveness of m4-8, m10-1, and m10-11 against WT PVMV Tn was evaluated in tomato SV-055 plants in a greenhouse. Across three distinct experiments, 49 plants pre-inoculated with m4-8, 23 with m10-1, and 22 with m10-11 were tested. Two weeks after inoculation with mild viruses, treated plants were challenged with PVMV Tn. Virus accumulation in test plants was monitored using indirect ELISA. Forty-six mock-inoculated tomato plants were inoculated with PVMV Tn for comparison. As expected, no apparent symptoms were observed in any of the tomato plants tested (Figure 3A). The accumulation of mild viruses, such as m4-8, in test tomato plants was significantly lower than that of PVMV Tn during the testing period (Figure 3B). Bioassays were performed on *C. quinoa* to clarify infections with WT or mild viruses. Results indicated that 44 of 46 mock control plants were infected with PVMV Tn, while only 1 of 49 m4-8-treated tomato plants was infected with PVMV Tn, and the protective efficacy was as high as 97.9%. In comparison, 4 of 23 m10-1 and 4 of 22 m10-11-treated tomato plants were infected with PVMV Tn with protective efficacies of 81.8% and 81.0%, respectively (Table 2).

### 3.4. Time Required for m4-8 to Provide Protection in Tomato Plants

To assess the optimal time required to provide adequate protection, two-true leaf stage tomato SV-055 seedlings pre-inoculated with m4-8 for 3, 5, 7, 9, or 11 days were challenged with PVMV Tn. Eight plants per treatment were tested. The protective effect on all tested plants was analyzed 14 days after the challenge. Indirect ELISA with RAs PVMV-CP to detect virus accumulation and bioassay to examine PVMV Tn infection were performed as described above. The results showed that the protective efficacies of m4-8 treatment for 3, 5, 7, 9, and 11 days were 25%, 50%, 62.5%, 100%, and 100%, respectively (Table 3 and Appendix A). The 9-day treatment enabled m4-8 to provide effective protection for tomato plants.

### 3.5. Clarification of the Genome Sequences of Taiwanese PVMV Isolates Tn and Xs-1

The whole-genome sequences of PVMV Tn and Xs-1 were determined using Sanger sequencing and deposited in GenBank with accession numbers of OR355467 and OR355466, respectively. The genomic RNA lengths of PVMV Tn and Xs-1 are 9797 nt and 9796 nt, respectively, and both encode a 3074-aa polyprotein. Sequence analysis revealed that the genome sequence of PVMV Tn shared 83.7–98.3% nt identity and 89.8–98.2% aa identity, respectively, with those of other PVMV isolates (Appendix A). Based on genome sequence alignment, PVMV Tn is most closely related to DSMZ-PV0257 (acc. no. MZ405642) isolated from Tabasco pepper (*Capsicum frutescens*) in Ghana. However, phylogenetic analysis showed that Tn and Xs-1 are closely related (Figure 4).

### 3.6. Comparison of Genome Sequences between PVMV-Tn and m4-8

The full-length genome sequence of m4-8 was also determined and compared with that of its WT PVMV Tn. Twenty nt substitutions were found that resulted in changes in three aa residues in P1, two in HC-Pro, two in P3, two in 6k1, two in CI, one in NIa-Pro, two in NIb, and four in CP. See Figure 5 for the details. In addition, the key VSR (HC-Pro) sequence of the mutant m4-8 was aligned with those of other PVMV isolates. None of the two mutated residues, K_117_E and K_282_R, were found in the HC-Pro of the analyzed viruses, except for K_282_R existing in Xs-1 (Appendix A).

## 4. Discussion

Cross-protection is a promising strategy for the control of plant virus diseases. Attenuated virus strains derived from aggressive viruses have been extensively used to prevent super-infection in economically important crops. For tomato crops, the attenuated strains of cucumber mosaic virus (CMV) KO2 [39] and tobacco mosaic virus (TMV) L_11_ and L_11_A in Japan [40], and pepino mosaic virus (PepMV) Sp13 and PS5 in Spain [41] have been practically applied to protect tomatoes. Currently, mild PepMV strains have been approved and commercialized as biopesticides, so-called vaccines, for tomato protection in Europe [41,42,43]. In this study, we established novel mild strains against PVMV in tomato crops.

RNA viruses have a high spontaneous error rate in replication, proposed as mutation rates of 10^−6^~10^−4^, that results in a virus species existing as a diverse population of genome mutants, including attenuated viruses in nature [44]. However, the selection of naturally attenuated mutants of RNA viruses has always been tricky due to the lack of available biological screening systems. The nitrite treatment and *C. quinoa* screening approach, previously exploited for obtaining attenuated PRSV [25] and ZYMV [33], contributes to this study to obtain protective PVMV mutants. In the previous study, the optimal condition of nitrite treatment for PRSV was recommended as the solution consisting of 0.4 M NaNO_2_ and 0.1 M NaOAc at pH 6.0, incubating at 20 °C for 30 min [25]. Using the same solution, the incubation temperature and time can affect the efficiency of nitrite-induced mutagenesis of different viruses, such as room temperature for 30 min for ZYMV [33] and 37 °C for 45 min for PVMV (this study). However, no correlation was found between nitrite treatment conditions and viral survival rates (Table 1).

Some previous studies have reported that *C. quinoa* is crucial for isolating attenuated viruses. *C. quinoa* is a local lesion-producing indicator plant and is also an effective tool commonly used in laboratories to isolate plant viruses [45]. Local lesion formation is a defensive hypersensitive response (HR) triggered by virus infection to localize and eliminate invasive viruses [46]. Previous studies revealed that protective mild viruses can infect but do not produce macroscopically observable HR on inoculated *C. quinoa* leaves [31,47,48,49], suggesting that viral virulence is associated with the induction of HR in *C. quinoa* leaves. To visualize invisible lesions, viruses were genetically engineered to carry the GFP reporter prior to mutagenesis [31,33]. However, infectious clones must be reconstructed to remove GFP for future application, which is inconvenient. To our knowledge, virus-triggered HR can be visualized as UV-excited fluorescence due to the accumulation of phenolic compounds [50]. We attempted to examine the virus-mediated fluorescence on *C. quinoa* leaves under UV illumination with an exciting wavelength of 365 nm using a Multigel-21 imaging system (TOPBIO, New Taipei City, Taiwan). Our results showed that, in addition to HR-induced lesions, fluorescence could be visualized in virus infection sites that do not induce HR. The PVMV mutants, m4-8, m10-1, and m10-11, exhibiting their weak and protective effects on *N. benthamiana* and tomato plants, were successfully obtained using this approach, picked from non-HR lesions. UV illumination can be used as an efficient method for selecting attenuated viruses from the leaves of local lesion-producing plants.

Ideal protective mild viruses should have no adverse effects on crops, be non-vector transmissible, and be genetically stable [25]. In addition, a zigzag pattern and low accumulation level are essential characteristics of protective mild viruses, which have been described in attenuated potyviruses, including East Asian Passiflora virus (EAPV) [51], Passiflora mottle virus (PaMoV) [49], PRSV [26], turnip mosaic virus (TuMV) [28,47], and ZYMV [31,33,52]. Our PVMV protective mutants also showed similar low accumulation patterns in *N. benthamiana* and tomato plants. The phenomenon is partially due to dysfunctional VSRs’ failure to counteract plant defense responses completely. Likewise, mild viruses must replicate at a basal level to maintain their cross-protection ability substantially [31].

Conserved motifs associated with multi-functions of potyviral HC-Pro have been addressed in previous studies. At the N-terminus, FWKG_7–10_ is involved in HC-Pro self-interaction [47], and KITC_49–53_ is essential for aphid transmissibility [53,54]. The central region of HC-Pro (aa positions 100–300) consists of two RNA-binding domains. The first domain is responsible for genome amplification and consists of four conserved motifs, FRNK_197_, KF(G)_153_, CDNQLD_214_, and IGN(R)_275–277_ [11]. The second domain, consisting of four conserved motifs, H(Y)HAKRFF_239_, GY_257_, PNC(G)_267_, and AIG_274_, plays a role in the suppression of PTGS. The motif CC_317–318_ is involved in systemic movement within host plants and synergism with other viruses [55,56]. HC-Pro has a protease domain at the C-terminal 156 aa consisting of six conserved motifs, C_369_, NIFLAML_372_, AELPRILVDH_427_, H_442_, LKAG(N)TV_457_, and VG_482_ at the active site [57]. The C-terminal PTK motif (310–312 aa) probably contributes to the binding of HC-Pro to the N-terminal DAG motif of CP [58,59]. The investigation of protective mild strains revealed that mutations of FRNK-to-FINK [31] and CDNQLD-to-CYNQLD [60] in the second RNA binding domain, M_117_-to-I, Q_121_-to-R, F_163_-to-C, and E_208_-to-K in the VSR region [33], and E_396_-to-N in the protease region [31] are associated with the pathogenicity and VSR of potyviruses. Compared with WT PVMV Tn, the HC-Pro of the protective PVMV mutant m4-8 contained two mutation sites, K_117_-to-E and K_282_-to-R, located in the VSR region. Furthermore, the analysis of the HC-Pro among different PVMV isolates suggested that the change in K_117_ to E may be critical (Appendix A). The K_117_E mutation involved in the VSR activity of PVMV HC-Pro will be investigated in the future. Likewise, m4-8 attenuation caused by other mutation sites cannot be ruled out.

Strain specificity may limit the protective effect of a mild virus in practice [24,25,26,61,62]. In our case, the other severe PVMV isolate, Xs-1, obtained from sweet pepper, was also used as a challenger to demonstrate the protective effect of m4-8 (Table 2 and Appendix A). The whole-genome sequences of Tn and Xs-1 were determined. Sequence analyses indicate that Tn is closely related to Xs-1, sharing 97.5% nt identity, and other PVMV isolates in China, Japan, and Ghana, sharing 97.8–98.3% nt identity (Appendix A). A total of 227 different nucleotides were found between the genome sequences of m4-8 and Xs-1. However, only one different residue in HC-Pro, K_117_E, was found between m4-8 and Xs-1. Multi-site mutations in the genome might facilitate the genetic stability and the protective broadness of m4-8. On the other hand, although some adverse effects might not be excluded entirely, some metabolites synthesized in protected plants are diverted to virus synthesis, and the yield of marketable fruits could be successfully preserved by treating protective viruses [63,64]. We also investigated tomato fruit yield between mock (healthy), PVMV-infected, and m4-8-protected plants under greenhouse conditions. Our results showed that although the fruit yield of m4-8-protected tomato plants was slightly affected compared to healthy plants, m4-8 significantly increased the fruit yield by comparison with the PVMV-infected tomato plants (Appendix A). It indicates that m4-8 is a satisfactory protector. In the protection assessment, it was observed that 1 out of the 49 m4-8-protected tomato plants became infected with PVMV Tn, as shown in Table 2. This may be due to escape from m4-8 protection or infection with resistance-breaking (RB) virus strains. The genetic makeup of individual plants may also affect the efficacy of protection. Employing a blend of various mild strains could potentially deter the emergence of RB strains.

In conclusion, this is the first study to obtain attenuated PVMV mutants using nitrite-induced mutagenesis. The PVMV mutant m4-8 derived from nitrite-induced mutagenesis has excellent potential as a protective agent against severe PVMV strains. The protective efficacy in the field and the insect vector transmissibility of m4-8 will be investigated in the future.

## Figures and Tables

**Figure 1 viruses-16-00687-f001:**
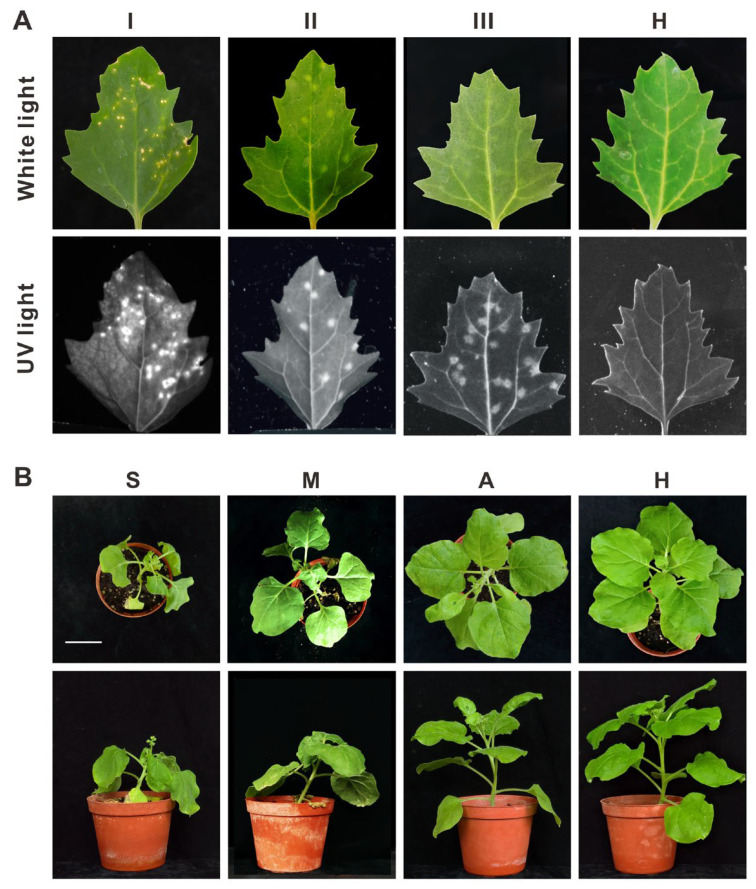
Responses of nitrite-treated pepper veinal mottle virus (PVMV) isolates on indicator plants. (**A**) Three local lesion types induced on inoculated *Chenopodium quinoa* leaves were observed and categorized as Types I–III. Type I represents typical necrotic spots, similar to those caused by the wild-type (WT) PVMV Tn. Type II represents chlorotic spots without necrosis. Type III represents inconspicuous blurred spots and could be easily observed under UV illumination (365 nm wavelength), appearing as bright as Type I and Type II spots. Local lesions were observed under white light (upper panel) and UV light (365 nm) (lower panel) at 10 days post-inoculation (dpi). (**B**) Type III PVMV isolates caused diverse levels of symptom severity in *Nicotiana benthamiana* plants. Those causing severe wilt symptoms, akin to those caused by WT PVMV Tn, were classified as severe mutants (S). Moderately severe mutants (M) caused leaf curling, mosaic, and stunting, but not wilting. Attenuated mutants (A) induced leaf curling and blistering. (H) represents healthy plants used for comparison. Photos were taken at 14 dpi and show top (**upper** panel) and side (**lower** panel) views. The scale bar shown is 5 cm.

**Figure 2 viruses-16-00687-f002:**
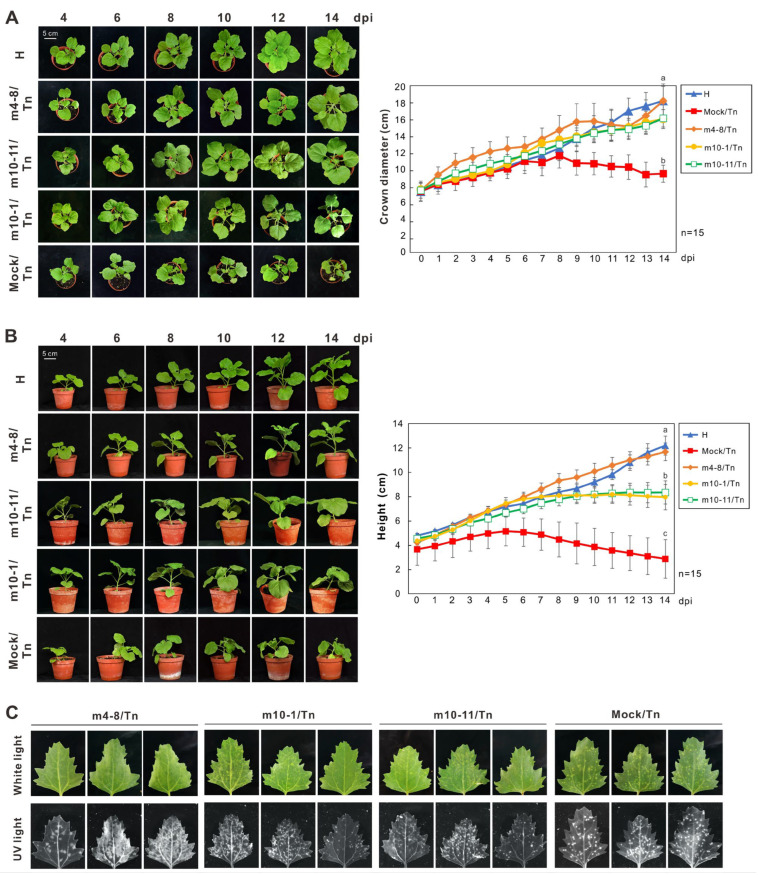
Evaluation of protective effects of the attenuated pepper veinal mottle virus (PVMV) mutants on *Nicotiana benthamiana* plants. Observation of symptom development and growth potential of test plants. Plants 4 days post-inoculation (dpi) with m4-8, m10-1, or m10-11 were challenged with wild-type PVMV Tn. Symptom development was recorded daily for 14 days. A total of 15 plants were used per treatment. The crown diameter (**A**) and height (**B**) of tested *N. benthamiana* plants are shown. The growth of a healthy plant, represented by H, is shown for comparison. The letters a, b, and c indicate significant differences between the two different treatments (**right** panel) (Mann–Whitney *U* test, *p* < 0.05) [37]. (**C**) Bioassays were performed on *Chenopodium quinoa* leaves to assess PVMV infection. Three leaves are shown for each treatment. Local lesions were observed under white light and UV light (365 nm) 10 days after the challenge. The appearance of necrotic spots on *C. quinoa* leaves indicates infection of PVMV Tn. UV illumination was performed to observe the infection of PVMV Tn and its mutants.

**Figure 3 viruses-16-00687-f003:**
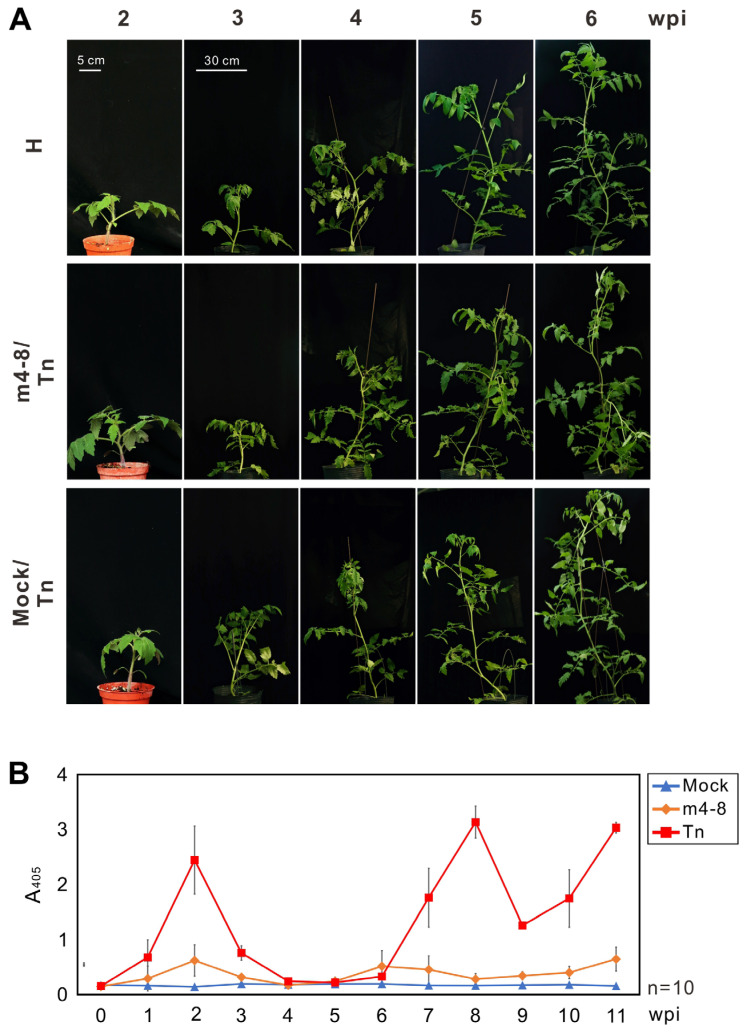
Evaluation of the protective effect of pepper veinal mottle virus (PVMV) mutant m4-8 in tomato plants. (**A**) The protected tomato plants with m4-8 were challenged with WT PVMV Tn at 2 weeks post-inoculation (wpi). Symptom development was observed for 6 weeks. (**B**) The accumulations of PVMV Tn and m4-8 were assayed by indirect enzyme-linked immunosorbent assay with the antiserum against the coat protein of PVMV. Upper uninoculated leaves of tomato plants were collected for assay. Crude sap of test plants 0–11 wpi at a 1/100 dilution was used for virus detection. Mock inoculation was used as the negative control.

**Figure 4 viruses-16-00687-f004:**
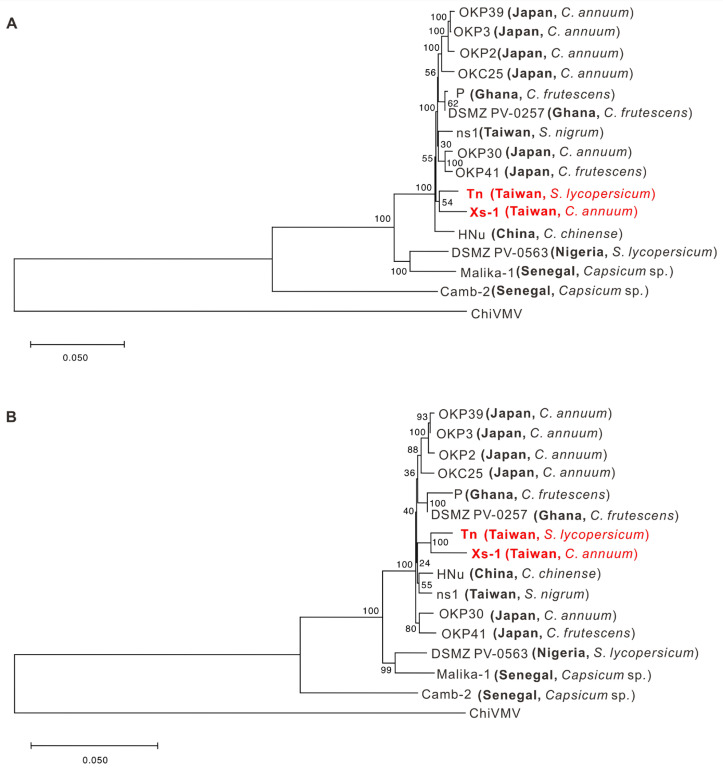
Phylogenetic relationships of different isolates of pepper veinal mottle virus (PVMV) based on nucleotide sequences of genome (**A**) and amino acid sequences of polyprotein (**B**). The dendrograms were produced using the neighbor-joining algorithm with 1000 bootstrap replicates. Scales refer to nucleotide or amino acid substitutions per site. Chilli veinal mottle virus (ChiVMV) was used as an outgroup. Accession numbers of viral sequences used for analysis are listed in Appendix A. The genome sequences of PVMV isolates determined in this study are indicated in red.

**Figure 5 viruses-16-00687-f005:**
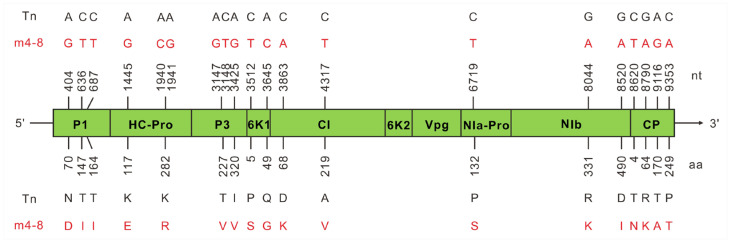
Schematic representation of genome sequence comparison between wild-type pepper veinal mottle virus (PVMV) Tn and m4-8. The positions of 20 nucleotides (nt) substitutions (**upper** panel) and 18 amino acids (aa) residue changes (**lower** panel) are indicated. The diverse residues within the m4-8 genome are indicated in red.

**Table 1 viruses-16-00687-t001:** Type and number of local lesions on *Chenopodium quinoa* leaves induced by pepper veinal mottle virus (PVMV) after nitrite treatment.

Experiment	Number of Local Lesions Induced by PVMV without Nitrite Treatment ^a^	Number of Local Lesions Induced by Nitrite-Treated PVMV ^b^				SR (%) ^c^	MR (%) ^d^
		Type I	Type II	Type III	Total No.		
1	463	9	3	0	12	2.6	0.6
2	685	147	13	0	160	23.4	1.9
3	436	196	0	0	196	45.0	0.0
4	634	165	34	1	200	31.5	5.5
5	856	28	0	0	28	3.3	0.0
6	657	53	10	0	63	9.3	1.5
7	804	602	27	0	629	78.2	3.4
8	763	122	8	0	130	17.0	1.0
9	965	20	289	0	309	32.0	29.9
10	778	0	31	11	42	5.4	5.4
Sum	7041	1342	415	12	1769	25.1	6.1

^a^ The number of local lesions was calculated based on 14 inoculated leaves. ^b^ Local lesions formed after nitrite treatment are categorized as Type I, necrotic spots, similar to those induced by wild-type PVMV Tn; Type II, chlorotic spots without necrosis; and Type III, inconspicuous blurred spots. ^c^ Survival rate (SR, %) = (total number of local lesions after nitrite treatment/total number of local lesions without nitrite treatment) × 100% [25]. ^d^ Mutation rate (MR, %) = (number of Type II and III local lesions/total number of local lesions without nitrite treatment) × 100% [38].

**Table 2 viruses-16-00687-t002:** Protective effectiveness of m4-8, m10-1, and m10-11 against aggressive PVMV strains (Tn and Xs-1) in *Nicotiana benthamiana* and tomato (*Solanum lycopersicum*) plants.

Treatment (Protector/Challenger)	Total Number of Test Plants	Number of Test Plants Infected with Tn or Xs-1 ^a^	Infection Rate of Tn or Xs-1 (IR, %) ^b^	Protective Efficacy (PE, %) ^c^
*N. benthamiana*				
m4-8/Tn	15	0	0.0	100.0
m10-1/Tn	15	8	53.3	46.7
m10-11/Tn	15	8	53.3	46.7
Mock/Tn	15	15	100.0	–
m4-8/Xs-1	10	0	0	100
Mock/Xs-1	10	10	100	–
*S. lycopersicum*				
m4-8/Tn	49	1	2.0	97.9
m10-1/Tn	23	4	17.4	81.8
m10-11/Tn	22	4	18.2	81.0
Mock/Tn	46	44	95.7	–

^a^ The infection of aggressive PVMV strains (Tn and Xs-1) was determined using bioassay on *Chenopodium quinoa* leaves showing necrotic spots and positive ELISA results. ^b^ IR (%) = (number of aggressive virus-infected plants/total number of test plants) × 100. ^c^ PE (%) = ([IR in the untreated population (IRU) − IR in the protected population (IRP)]/IRU) × 100.

**Table 3 viruses-16-00687-t003:** Protective effects of m4-8 on wild-type PVMV Tn in tomato plants at different protection times.

Treatment	Protection Days	Total Number of Test Plants	Number of Tn-Infected Plants ^a^	Infection Rate of Tn (IR, %) ^b^	Protective Efficacy (PE, %) ^c^
m4-8/Tn	3	8	6	75	25
	5	8	4	50	50
	7	8	3	37.5	62.5
	9	8	0	0	100
	11	8	0	0	100
Mock/Tn	–	8	8	100	–

^a^ PVMV Tn infection was determined using bioassay on *Chenopodium quinoa* leaves showing necrotic spots and positive ELISA results. ^b^ IR (%) = (number of PVMV Tn-infected plants/total number of test plants) × 100. ^c^ PE (%) = ([IR in the untreated population (IRU) − IR in the protected population (IRP)]/IRU) × 100.

## Data Availability

The genomic sequences of PVMV Tn and Xs-1 have been deposited in GenBank.

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
