# Peer review of "Development of Attenuated Viruses for Effective Protection against Pepper Veinal Mottle Virus in Tomato Crops"

_viruses, 2024, doi:10.3390/v16050687_

Round 1
Reviewer 1 Report
Comments and Suggestions for Authors
This manuscript introduces the development of attenuated viruses for effective protection against pepper veinal mottle virus (PVMV) in tomato. This manuscript is generally well written and well summarized. The result is great value for the development of attenuated virus, not only PVMV, but also the other plant virus. On the other hand, this manuscript needs the following some revisions before publication. After those revision, I agree that this manuscript will be published in this journal, Viruses.
Minor comments:
1) The font and size in some paragraphs are different, so please check them carefully. (ex. lines 29-48)
2) Line 98: Please write the centrifugal force in x g, not rpm.
3) Line 385: Delete Gho et al. 2023.
4) In protective effectiveness test, although m4-8 showed highly protective efficacy against Tn, a tomato plant had been infected with Tn. The authors should discuss whether this phenomenon is caused by resistance breaking or whether it is also caused by other factors.
Author Response
- This manuscript introduces the development of attenuated viruses for effective protection against pepper veinal mottle virus (PVMV) in tomato. This manuscript is generally well written and well summarized. The result is great value for the development of attenuated virus, not only PVMV, but also the other plant virus. On the other hand, this manuscript needs the following some revisions before publication. After those revision, I agree that this manuscript will be published in this journal, Viruses.
Response: Many thanks to the reviewer for the affirmation.
- The font and size in some paragraphs are different, so please check them carefully. (ex. lines 29-48)
Response: The font and size of the manuscript have been checked.
- Line 98: Please write the centrifugal force in x g, not rpm.
Response: 8000 rpm has been modified to 7155 ×g (L131).
- Line 385: Delete Gho et al. 2023.
Response: Deleted (L540).
- In protective effectiveness test, although m4-8 showed highly protective efficacy against Tn, a tomato plant had been infected with Tn. The authors should discuss whether this phenomenon is caused by resistance breaking or whether it is also caused by other factors.
Response: The statement “In the protection assessment, it was observed that one out of the 49 m4-8-protected tomato plants became infected with PVMV Tn, as shown in Table 2. This may be due to escape from m4-8 protection or infection with resistance-breaking (RB) virus strains. The genetic makeup of individual plants may also affect the efficacy of protection. Employing a blend of various mild strains could potentially deter the emergence of RB strains.” has been added to the Discussion (L566-571).
Reviewer 2 Report
Comments and Suggestions for Authors
To the authors - Very well done.
My only comment is that certain parts of Results could have been placed in the M&M - but this did not detract from the overall presentation. Figure 2 photos are a bit small and may deserve an entire page in the published article.
Author Response
- My only comment is that certain parts of Results could have been placed in the M&M - but this did not detract from the overall presentation.
Response: The corresponding descriptions of M&M are retained in the Results to help readers understand.
- Figure 2 photos are a bit small and may deserve an entire page in the published article.
Response: The photos in Figure 2 have been enlarged.